# Motion sickness symptoms during jumping exercise on a short-arm centrifuge

**Timo Frett**[1]*, **David Andrew Green**[2,3,4], **Michael Arz**[1], **Alexandra Noppe**[1], **Guido Petrat**[1], **Andreas Kramer**[5], **Jakob Kuemmel**[5], **Uwe Tegtbur**[6], **Jens Jordan**[1,7]

**1** Institute of Aerospace Medicine, German Aerospace Center, Cologne, Germany, **2** Space Medicine Team (HRE-OM), European Astronaut Centre, European Space Agency, Cologne, Germany, **3** KBRwyle GmbH, Cologne, Germany, **4** King's College London, London, United Kingdom, **5** Institute for Sport Sciences, University Konstanz, Konstanz, Germany, **6** Institutes of Sports Medicine, Hannover Medical School, Hannover, Germany, **7** Chair of Aerospace Medicine, University of Cologne, Cologne, Germany

* Timo.frett@dlr.de

## Abstract

Artificial gravity elicited through short-arm human centrifugation combined with physical exercise, such as jumping, is promising in maintaining health and performance during space travel. However, motion sickness symptoms could limit the tolerability of the approach. Therefore, we determined the feasibility and tolerability, particularly occurrence of motion sickness symptoms, during reactive jumping exercises on a short-arm centrifuge. In 15 healthy men, we assessed motion sickness induced by jumping exercises during short-arm centrifugation at constant $+1 Gz$ or randomized variable $+0.5$, $+0.75$, $+1$, $+1.25$ and $+1.5$ $Gz$ along the body axis referenced to center of mass. Jumping in the upright position served as control intervention. Test sessions were conducted on separate days in a randomized and cross-over fashion. All participants tolerated jumping exercises against terrestrial gravity and on the short-arm centrifuge during 1 $Gz$ or variable $Gz$ at the center of mass without disabling motion sickness symptoms. While head movements markedly differed, motion sickness scores were only modestly increased with jumping on the short-arm centrifuge compared with vertical jumps. Our study demonstrates that repetitive jumping exercises are feasible and tolerable during short-arm centrifugation. Since jumping exercises maintain muscle and bone mass, our study enables further development of exercise countermeasures in artificial gravity.

## Introduction

Lack of terrestrial gravity during space travel produces multiple physiological adaptations challenging astronaut performance and health. The issue is particularly relevant for future deep space missions. Countermeasures relying on strength and endurance exercises help maintaining skeletal muscle [1] and cardiopulmonary fitness [2]. Current exercise countermeasures on the International Space Station are individually tailored for each astronaut. In general, an integrated resistance and aerobic training schedule is prescribed [3–5]. Crewmembers typically exercise six days per week, which consumes significant crew time and resources [6,7]. Yet,

**Funding:** The study was part of the German National Centrifuge Program 2 financed by the Space Administration of the German Aerospace Center. KBRwyle GmbH provided the salary for D.G but did not have any additional role in the study design, data collection and analysis, decision to publish, or preparation of the manuscript.

**Competing interests:** KBRwyle GmbH provided the salary for D.G but did not have any additional role in the study design, data collection and analysis, decision to publish, or preparation of the manuscript. The authors declare no competing interests as KBRwyle GmbH had no role in the study design and thus this does not alter our adherence to PLOS ONE policies on sharing data and materials.

with current countermeasures, lower limb bone mass and muscle volume was still reduced after 16–28 weeks in space [8]. Moreover, countermeasures for potentially serious changes in ocular and brain structures likely resulting from chronic cephalad fluid shifts, the so called space associated neuro-ocular syndrome [9, 10], have not been established. Other approaches such as passive axial loading suits or lower body negative pressure systems [11–13] affect only parts of the complex physiological adaption process during long-term space missions.

Artificial gravity elicited through axial acceleration on short-arm human centrifuges, which distributes fluids to the lower part of the body, has been developed as potential countermeasure. Centrifugation may also help maintaining coordination and vestibular function, which are crucial when arriving on other celestial bodies. Yet, centrifugation when simply added to current countermeasures may not be practical given the tight schedule of astronauts. Combined centrifugation and physical exercise may be more efficient. Because reactive jumps appear to maintain skeletal muscle as well as bone mass in bed rest [5], jumping exercises during centrifugation are particularly promising. However, exercise-induced head movements within a rotating environment can produce severe motion sickness symptoms or illusory sensations through cross-coupled angular accelerations of semi-circular canals [6]. The issue is complicated by the steep g gradient away from the rotation axis during short-arm centrifugation [7]. Leg press exercises were tolerated during centrifugation, however, subjects were restrained to avoid head movements [8]. Therefore, the aim of our study was to determine the feasibility and tolerability of reactive jumping exercises during short-arm human centrifugation.

## Methods

### Study participants

We included 15 healthy men (26.4 ± 5.8 yrs; 180.9 ± 4.0 cm; 77.2 ± 5.8 kg) who were naïve to jumping exercises during centrifugation. Prior to the study, participants completed a brief medical questionnaire detailing their drug and medical history and passed a standardized centrifuge medical screening that includes clinical-chemical analyses of blood and urine, stress electrocardiogram, and orthostatic testing. Participants were excluded if they were in pain, or had any significant current or history of musculoskeletal, cardiovascular or neurological disorder or injury that could affect the ability to perform exercise. All participants were recreationally active (engaging in a minimum of two sport sessions per week) in order to facilitate exercise performance and minimize risk of injury during centrifugation. All participants gave written informed consent to participate in the study. The study was approved by the North Rhine ethical committee (Ref: 2017122).

### Protocol

Participants attended to the laboratory at: envihab (DLR, Cologne, Germany) on four testing sessions separated by at least three resting days to allow for muscle recovery. In a fifth session participants ran on a treadmill at the German Sports University in Cologne. Participants were not permitted to take anti-emetic medication (i.e. scopolamine) and were offered light food (bananas, cereal bars) and non-sparkling water during each protocol to ensure hydration and glycaemia. Our experiment on motion sickness was part of a broader physiological investigation of jumping exercises during centrifugation that will be published elsewhere. Briefly, we compared effects of jumping exercises in the supine position on a short-arm centrifuge during spinning at different gravity level with jumping in upright position in terrestrial gravity (see Table 1).

**Table 1. Exercise conditions for each participant.**

| Condition | Description |
|---|---|
| Terrestrial Gravity | 15 x 15 vertical jumps in terrestrial gravity |
| Continuous AG | 15 x 15 jumps at constant +1 $Gz^*$ at CoM |
| Variable AG | SAHC: 3 x 15 jumps at +0.5 $Gz^*$ |
| | 3 x 15 jumps at +0.75 $Gz^*$ |
| | 3 x 15 jumps at +1 $Gz^*$ |
| | 3 x 15 jumps at +1.25 $Gz^*$ |
| | 3 x 15 jumps at +1.5 $Gz^*$ |
| | in randomized order |

$^*$The value refers to $Gz$ at the center of mass

Prior to recording, participants were familiarized with equipment and testing procedures including a brief centrifugation run. In two testing sessions, subjects performed jumping exercises in artificial gravity (AG) on the DLR-short-arm centrifuge at constant +1 $Gz$ along the subject´s body axis (Continuous AG) and with +0.5, +0.75, +1, +1.25 and +1.5 $Gz$ along the subject´s body axis in randomized order (Variable AG). Jumping in the upright position against terrestrial gravity served as control intervention (Terrestrial gravity). The study was conducted in a randomized controlled cross-over fashion.

Participants performed jumping exercises in the supine position on the short-arm centrifuge using a horizontal sledge (Figs 1 and 2) against a fixed footplate. The jumping sledge was attached to the short-arm centrifuge via low friction bearings that by riding along rails permitted linear movements along the centrifuge arm (Fig 3A). In addition, the sledge allowed for pitch at participants´ center of mass to facilitate natural jumping movements (Fig 1).

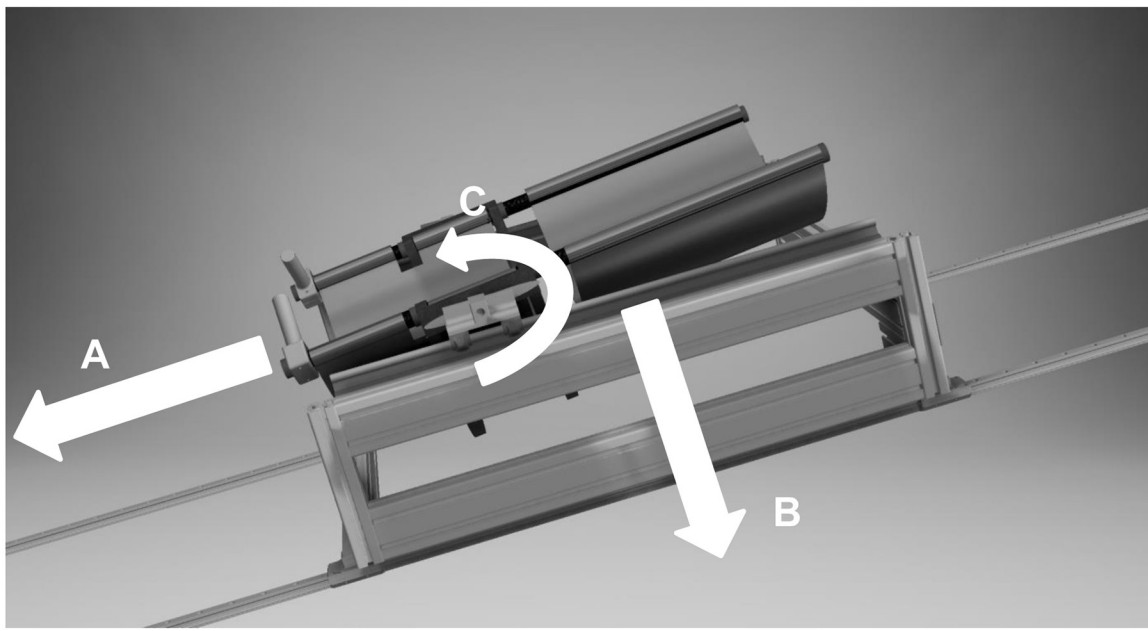

**Fig 1. Schematic of the jumping sledge used on the short-arm human centrifuge.** Participants were secured in supine position with a safety belt controlling their movement using two hand grips while jumping against a footplate mounted to the centrifuge. Due to the sledge design, movements along the centrifuge radius (A) against earth´s gravity (B) and in pitch axis around the center of mass (C) are possible.

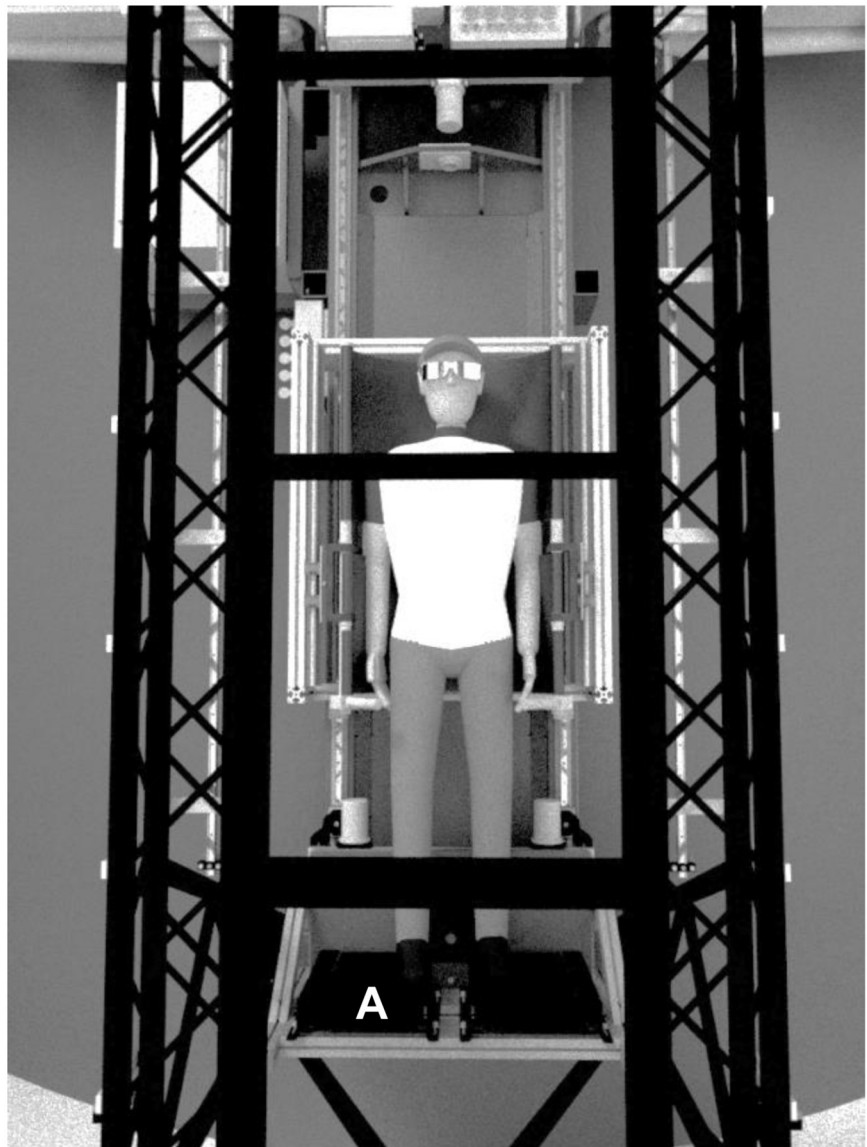

**Fig 2. Presentation of participants position on the short-arm human centrifuge in bird's-eye perspective.** During centrifugation participants performed jumping exercises against a footplate (A).

Participants were fastened on the sledge by safety belts around hip. The head was not restrained. Each centrifugation session lasted approximately 30 min. In protocol 2, each $G$–level lasted for around 6 min. Onset and offset acceleration of the centrifuge were 0.1 $G$/sec. We terminated centrifugation when participants demonstrated pre-syncopal signs or symptoms.

## Data acquisition and analysis

During centrifugation, five lead electrocardiogram, brachial cuff blood pressure, finger pulse oximetry (Philipps IntelliVue®), and a live video feed were continuously monitored subjects by an experienced physician.

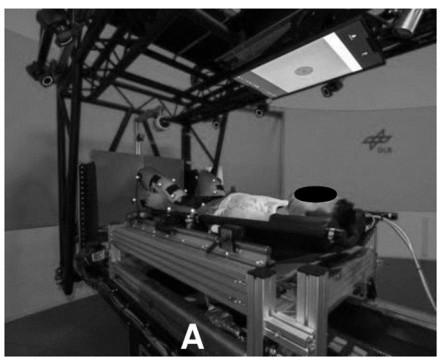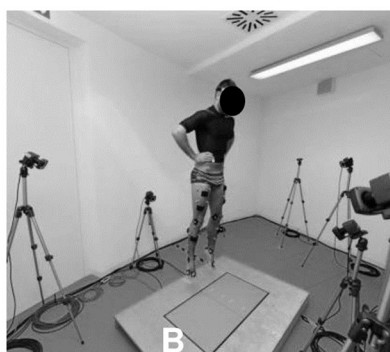

**Fig 3. Participant´s jumping position during (A) continuous or variable centrifugation on the short-arm human centrifuge and (B) vertically against terrestrial gravity.**

We assessed tri-axial (pitch, yaw, and roll) head movement velocities throughout using a wearable inertial sensor (Shimmer3, Shimmer, Dublin, Ireland) secured with an elasticated band on the forehead. We determined motion sickness susceptibility prior to the study using the Motion Sickness Susceptibility Questionnaire (MSSQ) short-form [10] that yields MSA (based on childhood experience (before age 12) and MSB for that over the last 10 years (max score = 54).

Directly before, and immediately following each condition, participants completed Subjective Motion Sickness Rating, Motion Sickness Assessment Questionnaire (MSAQ), Positive and Negative Affect Schedule (PANAS), and Epworth Sleepiness Scale (ESS) questionnaires. Subjective Motion Sickness Rating's range from 0 "I am feeling fine" to 20 "I am about to vomit" [14]. The MSAQ was used to measure (1 to 9 max) various dimensions (e.g. gastrointestinal, sopite) of motion sickness [15]. PANAS was used to measure the effect of symptoms induced by jumping upon mood. Participants rated each item on a Likert scale from 1 "not at all" to 5 "very much". The ESS (which via rating from 0 (non-) to 3 "high chance of dozing" in 8 contexts) since "drowsiness" is a cardinal symptom of motion sickness [16–18]

In addition, participants were asked regularly during centrifugation whether they were experiencing any motion sickness symptoms, and to report any unexpected symptoms such as tunnel vision or tumbling sensations.

During centrifugation five lead ECG (Philipps IntelliVue®), cuff blood pressure and $SpO_2$ as well as a live video feed were used to continuously monitor subjects by an experienced physician. Any run where participants demonstrated pre-syncopal symptoms was terminated immediately.

## Statistical analysis

Mean head movement (Pitch, Yaw, Roll) velocities were compared between jumping sessions 1–15 for each condition using analysis of variance with repeated measurement. All questionnaire pre and post data was compared between conditions per participant. Pre-data represented scoring from every questionnaire before starting of the individual condition and post-data for every questionnaire after completion of each condition. Non-parametric tests (Friedman´s Chi-Square) were performed to evaluate whether there was an effect of condition. If significant differences across conditions were observed, post-hoc tests with pairwise comparisons using Dunn-Bonferroni were performed to determine which condition was significant different.

All statistical tests were conducted using SPSS version 21 (IBM Corp., USA) with $\alpha < 0.05$ indicating significance.

## Results

All participants tolerated well jumping exercises against terrestrial gravity and on the short-arm centrifuge during both, the continuous and the variable centrifugation protocol. Only one subject experienced presyncopal symptoms requiring termination of the Variable AG protocol but completed all other protocols without similar symptoms. No disabling motion sickness symptoms occurred that required termination of testing. Serious adverse events did not occur.

Mean head movement velocities in pitch axes did not differ between centrifugation protocols (Fig 4) but compared to terrestrial condition (p = 0.000, dfs = 14). In the eccentric phase of the jumps, mean positive peak pitch angular velocity (Fig 5) was significantly greater during continuous (t (14) = 5.06, p < 0.001) and variable centrifugation (t (14) = 6.27, p < 0.001) compared to the terrestrial control condition. During concentric movements against the centrifuge´s gravity vector, mean negative pitch angular velocity was also significantly greater in continuous (t (14) = -8.503, p < 0.001) and variable centrifugation protocols (t (14) = -3.055, p = 0.009) compared with the control intervention. We observed no significant changes in head movements across time, F = 0.827, p = 0.643, partial $\eta 2$ = 0.045, n = 15 (Greenhouse-Geisser).

No participant reported motion sickness before the training sessions commenced. Motion Sickness Susceptibility (MSSQ) scores were 10.84 ± 4.52 with sub-scores for MSA (5.68 ± 2.70) and MSB (5.37 ± 2.93).

After the interventions, Subjective Motion Sickness Ratings were low with 1.33±0.48 following Terrestrial gravity intervention, 2.53 ± 1.45 following Continuous AG, and 2.15 ± 1.14 following Variable AG. Post-hoc analysis (Dunn-Bonferroni) across conditions showed that

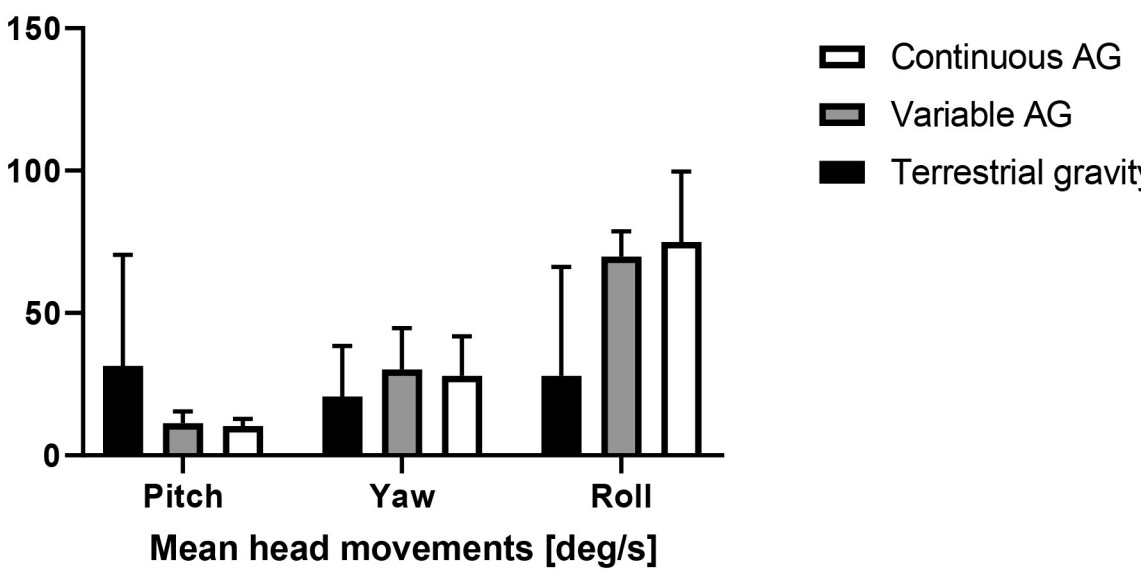

**Fig 4. Mean (± SD) head movement velocities in roll, yaw and pitch for each condition.**

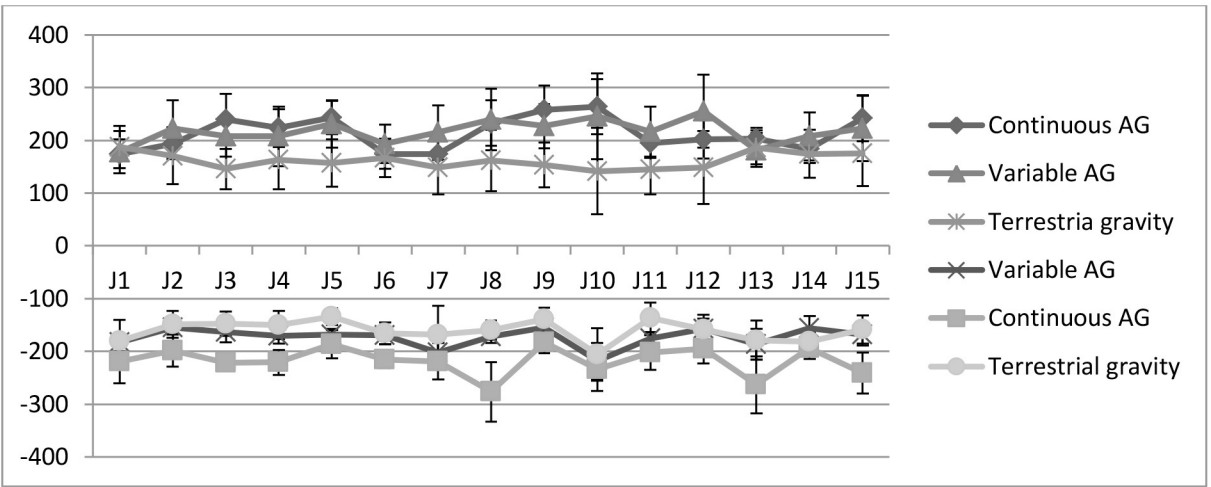

**Fig 5. Mean (± SD) peak pitch angular velocities during each jumping session and in each condition.** Subsequent jumps are labeled as J1 to J 15.

Subjective Motion Sickness ratings were significantly higher during continuous centrifugation compared to terrestrial control condition (z = 2.527, p = 0.034).

Post condition mean Motion Sickness Assessment Questionnaire scores were relatively low (Fig 6) and did not differ between conditions (Friedman´s Chi-Square $\chi^2$ (2) = 0.792, p = 0.673).

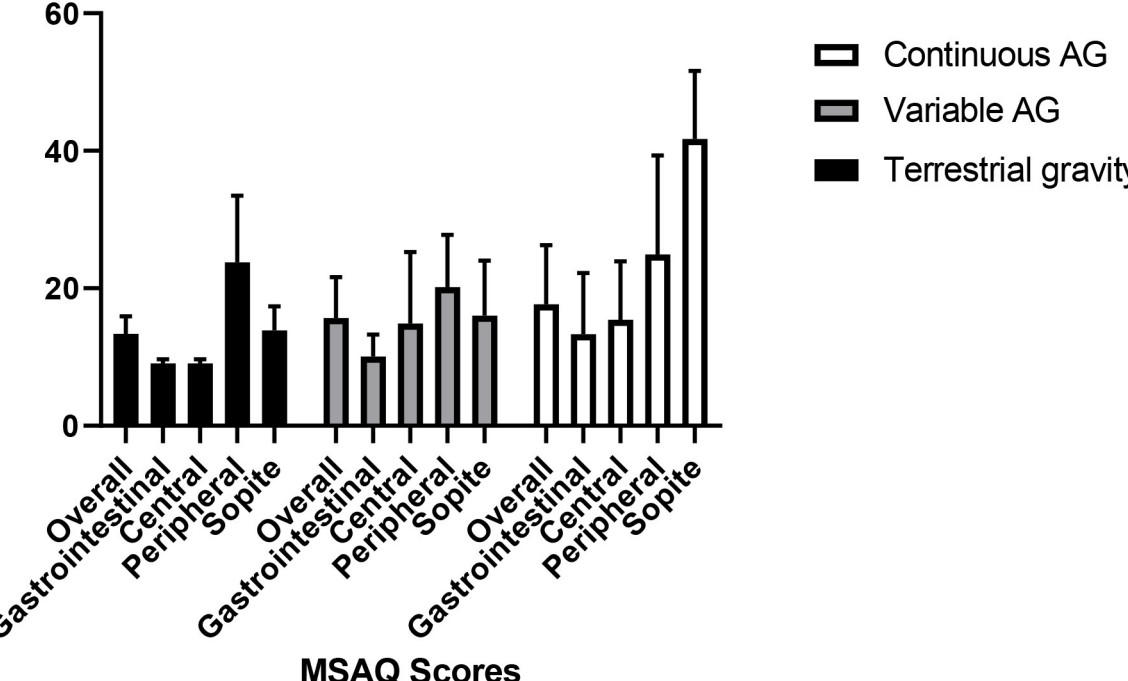

**Fig 6. Motion sickness scoring from MSAQ questionnaire for each condition.**

Post condition PANAS scores tended to be more positive in all conditions with PA: 26.25 ± 9.63, NA: 13.23 ± 4.81 following Continuous AG, PA: 29.76 ± 10.00, NA: 10.61 ± 1.66 following Variable AG and PA: 23.26 ± 8.72, NA: 15.73 ± 5.31 following terrestrial control condition,. Between the terrestrial control condition and both centrifuge conditions no significant effect (PANAS P $\chi^2$ (2) = 5.636, p = 0.060, $\chi^2$ (2) = 4.769, p = 0.092) occurred.

Post condition Epworth Sleepiness Scale scores did not differ significantly between conditions ratings were numerically slightly higher with centrifugation (Continuous AG: 8.28 ± 5.31, Variable AG: 8.07 ± 5.89, terrestrial control condition: 6.33 ± 3.67).

## Discussion

Our study demonstrates that repetitive voluntary jumping exercises are both feasible and tolerable during short-arm centrifugation at levels ranging from +0.5 to +1.5 *Gz* at the center of mass along the body axis. Indeed, the intervention was well tolerated by recreationally fit individuals who were naïve to jumping exercises during centrifugation as long as they were briefly familiarized. Study participants could move their heads freely within certain safety limits on the centrifuge and perform jumping exercises without experience increased motion sickness levels. Thus, contrary to the common perception that whole-body movements, including head motion during short-arm centrifugation result in motion sickness and related symptoms *per se*, we demonstrated that vigorous repetitive jumping is possible without induction of negative motion sickness symptoms.

Head movements within a rotating environment produce cross-coupled angular accelerations in the semicircular canals. The mechanism can trigger adverse vestibular stimulation with symptoms ranging from mild discomfort (e.g. sweaty palms) to severe nausea, vomiting or even loss of consciousness [6]. Yet, not only was repetitive jumping possible but no participant needed to drop out due to motion sickness symptoms. In fact, Motion Sickness Scores and Motion Sickness symptoms were low in all conditions. The finding is remarkable given the high values for head yaw, pitch, and roll velocities being generated in all conditions that are excess of those previously defined as being associated with comfort zones [9]. Moreover, the comparison between both centrifuge conditions reveals the interesting fact that alternating gravity levels seems to have only minor effects on the increase of motion sickness scoring or other related symptoms.

Our study extends the recent findings of Piotrowski et al [8] who demonstrated that leg press exercises on a sledge during centrifugation albeit with head movement restraint, could be tolerated. Thus, contrary to that previously thought rapid, forceful and complex voluntary repetitive movement such as jumping can be implemented during short-arm centrifugation. The cardiovascular burden imposed by short-arm centrifugation may promote presyncopal symptoms that can progress to frank syncope. The fact, that only one presyncopal event occurred during the Variable AG condition is reassuring. It is likely that jumping or squat exercise during centrifugation can help to maintain orthostatic tolerance even in a steep +*Gz* gravity gradient.

PANAS Negative Affect (NA) Scores tended to be slightly lower during centrifugation. These findings, albeit non-significant may be explained by participants perceiving centrifugation as exciting–particularly for unexperienced participants.

The fact that only men were included is a limitation that was part of the study design in which our experiment was included. In our study, both average MSA and MSB MSSQ scores were relatively low compared to normative populations [10,11]. Thus, whether similar results would be observed in more or highly sensitive individuals is unknown. While some subjects in our study scored relatively high in terms of motion sickness sensitivity (MSB > 11), none

featured motion sickness requiring test termination. While the issue warrants further study, astronaut populations undergo tight medical screening and are not likely to have high motion sickness susceptibility. We cannot exclude that repeated exposure as part of a countermeasure protocol mitigates motion sickness symptoms completely. Since our study only included men, our findings cannot be simply extrapolated to women. Indeed, previous studies reported impaired vasoconstriction leading to impaired orthostatic tolerance in women after bed rest [12].

Despite these issues, we suggest that jumping exercises on a short-arm centrifuge are not generally restricted by disabling motion sickness symptoms. We speculate that being 'in control' may have increased the tolerability against cross-coupled effects during head movements while exercising on the short-arm centrifuge. This could be explained with increased controllability of the unknown setting on a centrifuge [13]

Since jumping exercise have been proven efficient in maintaining bone and muscle mass, our study enables further development of exercise countermeasures in Artificial Gravity.

## Supporting information

**S1 File.**
(RAR)

## Acknowledgments

The authors would like to thank the DLR centrifuge team and DLR study team for the conduction of the experiment. Further gratitude belongs to the principle investigator teams from University of Konstanz and the German Sport University for their cooperation.

## Author Contributions

**Conceptualization:** Timo Frett.

**Data curation:** Michael Arz.

**Formal analysis:** Timo Frett, David Andrew Green.

**Investigation:** Michael Arz, Alexandra Noppe, Guido Petrat, Jakob Kuemmel.

**Project administration:** Jens Jordan.

**Software:** Michael Arz.

**Supervision:** Uwe Tegtbur.

**Writing – original draft:** Timo Frett.

**Writing – review & editing:** Timo Frett, David Andrew Green, Alexandra Noppe, Andreas Kramer, Jens Jordan.

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
