## [Decision Letter · Decision Letter 0]

3 Apr 2020

PONE-D-20-07198

Motion sickness symptoms during jumping exercise on a short-arm centrifuge

PLOS ONE

Dear Mr. Frett,

Thank you for submitting your manuscript to PLOS ONE. After careful consideration, we feel that it has merit but does not fully meet PLOS ONE’s publication criteria as it currently stands. Therefore, we invite you to submit a revised version of the manuscript that addresses the points raised by a reviewer and the editorial comments noted below during the review process.

We would appreciate receiving your revised manuscript by May 18 2020 11:59PM. To enhance the reproducibility of your results, we recommend that if applicable you deposit your laboratory protocols in protocols.io, where a protocol can be assigned its own identifier (DOI) such that it can be cited independently in the future. For instructions see: http://journals.plos.org/plosone/s/submission-guidelines#loc-laboratory-protocols

We look forward to receiving your revised manuscript.

Kind regards,

Dr. Sakamuri V. Reddy

Academic Editor

PLOS ONE

Additional Editor Comments (if provided):

The manuscript show repetitive jumping exercises during centrifugation is feasible and tolerated. This would allow future studies to develop training protocols of exercise as countermeasures in artificial gravity. Physiological response of jumping exercises was analyzed, but not included. The authors should follow the journal format referring to an article published. For ex., Introduction should follow Methods (p.8-10) section. Introduction section (p.3) is written verify briefly. They could have provided more details on bone loss and nervous system affected with gravity changes/or under low gravity, ex., “However, bone mass is not maintained”; “…physical exercise may be more efficient.”

2. Please ensure you have closely checked your Short Title for any typographic errors (e.g. "Repetetive")

"The authors declare no competing interests."

We note that one or more of the authors are employed by a commercial company: KBRwyle GmbH.

Reviewers' comments:

Reviewer's Responses to Questions

**Comments to the Author**

1. Is the manuscript technically sound, and do the data support the conclusions?

Reviewer #1: No

2. Has the statistical analysis been performed appropriately and rigorously? 

Reviewer #1: N/A

3. Have the authors made all data underlying the findings in their manuscript fully available?

Reviewer #1: No

4. Is the manuscript presented in an intelligible fashion and written in standard English?

Reviewer #1: Yes

5. Review Comments to the Author

Reviewer #1: General points

The present study examined the tolerability of motion sickness by jumping exercise during artificial gravity in 15 healthy men.

The authors prepared five conditions, jumping against the force plate of the horizontal jump sledge at 0.5, 0.75, 1.0, 1.25, 1.5 G, and 1 G centrifuge as well as vertical jumping on the jumping board.

The result is there were no difference in motion sickness scores during various Gz and 1 G centrifuge as well as vertical jumping.

The conclusion is very simple, but there are several concerns from the viewpoint of 1) performance, 2) description of physics.

The scheme of jumping during acceleration is very difficult to read from the text. They should show the photos or drawings of the jumping performance.

Head movements during centrifuge are very difficult to understand from the text. The authors should show the photos or drawings, or the recordings of accelerometer of the head of the subject should be shown in relation with the three-dimensional gravitational acceleration. Fig. 3 is a transition in the time course, but the actual drawing of three-dimensional G transition is requested.

The unit of gravitational acceleration should be the capital G italic to distinguish it from the gram unit of mass, and a space is necessary between numeral and G.

The authors should use the +Gz and −Gz in reference to the body axis i.e. +Gz as the gravity of head to the foot, and −Gz as the foot to the head, of the subject, not in terrestrial +/−Gz.

Comma and period are mixed in the method of displaying the decimal point. It should be unified to either, and preferably to a period in the English sentence.

Motion sickness scores and angular accelerations of head movements should be expressed by graphs not tables.

Minor Points

Please show the rationale to separate the sessions at least three days.

As far as the reviewer know, the name of the facility is Envihab, not :envihab, did they change the name?

6. PLOS authors have the option to publish the peer review history of their article (what does this mean?). If published, this will include your full peer review and any attached files.

Reviewer #1: Yes: Satoshi Iwase, Department of Physiology, Aichi Medical University

---

## [Author Response · Author response to Decision Letter 0]

27 Apr 2020

Dear Reviewer,

thank you for your helpful feedback to our manuscript. We have change all points as requested and listed them with description in the "response to reviewer" document. We hope that the manuscript now meet your expectations. 

Kind regards,

Timo Frett

---

## [Decision Letter · Decision Letter 1]

26 May 2020

Motion sickness symptoms during jumping exercise on a short-arm centrifuge

PONE-D-20-07198R1

Dear Dr. Frett,

We are pleased to inform you that your manuscript has been judged scientifically suitable for publication and will be formally accepted for publication once it complies with all outstanding technical requirements.

With kind regards,

Dr. Sakamuri V. Reddy

Academic Editor

PLOS ONE

Additional Editor Comments (optional):

Reviewers' comments:

Reviewer's Responses to Questions

**Comments to the Author**

1. If the authors have adequately addressed your comments raised in a previous round of review and you feel that this manuscript is now acceptable for publication, you may indicate that here to bypass the “Comments to the Author” section, enter your conflict of interest statement in the “Confidential to Editor” section, and submit your "Accept" recommendation.

Reviewer #1: (No Response)

2. Is the manuscript technically sound, and do the data support the conclusions?

Reviewer #1: Yes

3. Has the statistical analysis been performed appropriately and rigorously? 

Reviewer #1: Yes

4. Have the authors made all data underlying the findings in their manuscript fully available?

Reviewer #1: Yes

5. Is the manuscript presented in an intelligible fashion and written in standard English?

Reviewer #1: Yes

6. Review Comments to the Author

Reviewer #1: The revised manuscript was well revised, and the reviewer agrees to accept it with only one minor addition, which refers to the relation of neck bending during rotation. It has been known that the neck flection or extension to the lateral direction or anterior flection/posterior extension might cause illusory sensation, sometimes nausea and vomiting. This illusory sensation by head-movement due to the Coriolis effect during rotation should be touch upon in the Introduction or Discussion section. [Clement G. et al.(2001) Eur J Appl Physiol 85: 539-45. And please refer to the first report on short arm centrifuge effects on human body [Iwase S (2005) Acta Astronautica 57: 75-80]

For Figures, schema and photo of the experimental device was excellent. The presentation in these forms are quite well appreciated.

Graph of Figure 5 was difficult to read due to the suppressed shape in top-bottom direction. Please extend to the top-bottom direction, and the shape is preferably portrait (not landscape) shape.

7. PLOS authors have the option to publish the peer review history of their article (what does this mean?). If published, this will include your full peer review and any attached files.

Reviewer #1: Yes: Satoshi Iwase

---

## [Editor Report · Acceptance letter]

28 May 2020

PONE-D-20-07198R1 

Motion sickness symptoms during jumping exercise on a short-arm centrifuge 

Dear Dr. Frett:

I am pleased to inform you that your manuscript has been deemed suitable for publication in PLOS ONE. Congratulations! Your manuscript is now with our production department. 

With kind regards,

on behalf of

Dr. Sakamuri V. Reddy 

Academic Editor

PLOS ONE